

# An assessment for health education and health promotion in chronic disease demonstration districts: a comparative study from Hunan Province, China

Qiaohua Xu, Biyun Chen, Donghui Jin, Li Yin and Yuelong Huang

Department of Chronic Diseases Control and Prevention, Hunan Provincial Center for Diseases Control and Prevention, Changsha, Hunan, China

## ABSTRACT

**Background**. Cost-effective strategies of chronic disease control, integrated health education and health promotion play important roles in the programs of chronic disease demonstration districts in China. The performance of these districts can be directly assessed by their health education and promotion work. However, there have been only a few performance assessments done on these programs, most of which made without the inclusion of proper quality indicators. This study was designed to establish a framework of indicators for outcome evaluation of health education and promotion efforts in Chinese districts, and explore the factors involved in promoting these efforts.
**Methods**. A modified two-round Delphi survey was first used to construct quality indicators on a nine-point Likert scale. With those indicators, the rank sum ratio (RSR) method was then conducted through rank conversion and parametric statistics, to assess and classify the performance of ten districts or counties randomly chosen both from demonstration and non-demonstration districts in the Hunan province.
**Results**. The Delphi process produced seven themes and 25 sub-themes as quality indicators. The seven themes included organizational management, financial support, professional personnel, health education and promotion, residents' health awareness and behaviors, residents' satisfaction, and residents' health literacy. The districts were classified into four levels by RSR as follows: One demonstration district at the first-ranked level, five other demonstration districts at the second-ranked level, all non-demonstration districts at the third-ranked level. None were at the fourth-qualified level.
**Discussion**. Chronic disease demonstration districts performed better on the work of health education and health promotion than the non-demonstration districts. The work should be focused on the following measures of chronic diseases: organizational management, financial support, media-related broadcasting, technical support, community-based promotion and supportive environment, and people's enhanced awareness and health literacy.

Corresponding author
Yuelong Huang, ylh410@126.com

# INTRODUCTION

For some time, chronic diseases, also known as non-communicable diseases (NCDs), have been the top health threat for Chinese people and now pose an increasing disease burden (*Bureau of Diseases Prevention and Control of the NHPFC, 2012*; *Huang, Yu & Koplan, 2014*). Currently, 260 million people nationwide are diagnosed with NCDs which are responsible for 85% of the mortality rate and 70% of the disease burden in the country (*The State Council Information Office of the People's Republic of China, 2012*). Thus, the National Health and Family Planning Commission of China (NHFPC) supported a program of NCD demonstration districts (or counties) in 2010, incorporating both important public health projects for Chinese medical health reform and a work plan of NCD control in China (2012–2015) (*The National Health and Family Planning Commission of China, 2016*). Since that time, a series of national or provincial NCD demonstration districts have been successively set up across the country. By 2017, thirty-four counties or districts were nominated as NCD demonstration districts in the Hunan province (located in central China), ten of which were nominated as national districts; the rest were nominated as provincial districts (pending approval by the NHFPC).

As cost-effective NCD control strategies (*Bayarsaikhan & Muiser, 2007*), health education and health promotion also play important roles in the program of NCD demonstration districts.

The status of the districts can be directly assessed by their health education and promotion. Nevertheless, considering both advanced strategies and work experiences of health education and promotion in developed countries (*Butler, 2001*; *Puska, 2008*; *Daniel et al., 1999*), the implementation of these strategies and this work has had a late start in China. To date, very few systematic assessments have been conducted, especially in NCD demonstration districts. The biggest challenge has been to find proper quality indicators for these assessments. Without such indicators, it is unclear how well these districts are able to respond to the significant challenges of health education and promotion in their region.

Thus, our study was the first designed to develop quality indicators by the modified RAND/UCLA Delphi method originating from Kathryn Fitch (*Fitch et al., 2001*). With these indicators, we were able to get acquainted with the situation of the districts by comparatively assessing the performances between NCD demonstration districts and non-demonstration districts through the rank sum ratio (RSR) technique (*Wang et al., 2015b*), and discover important factors relevant to further the progress of health education and promotion work.

# MATERIALS AND METHODS

## Study design

As mentioned above, a modified Delphi combined with the RSR technique was comprehensively used as an evaluation tool and method for this study. Figure 1 shows the overall flow diagram of this study. The Delphi procedure was firstly used to build quality indicators for evaluation of health education and promotion in NCD demonstration districts. The detailed procedure was conducted using the following steps (1–5):
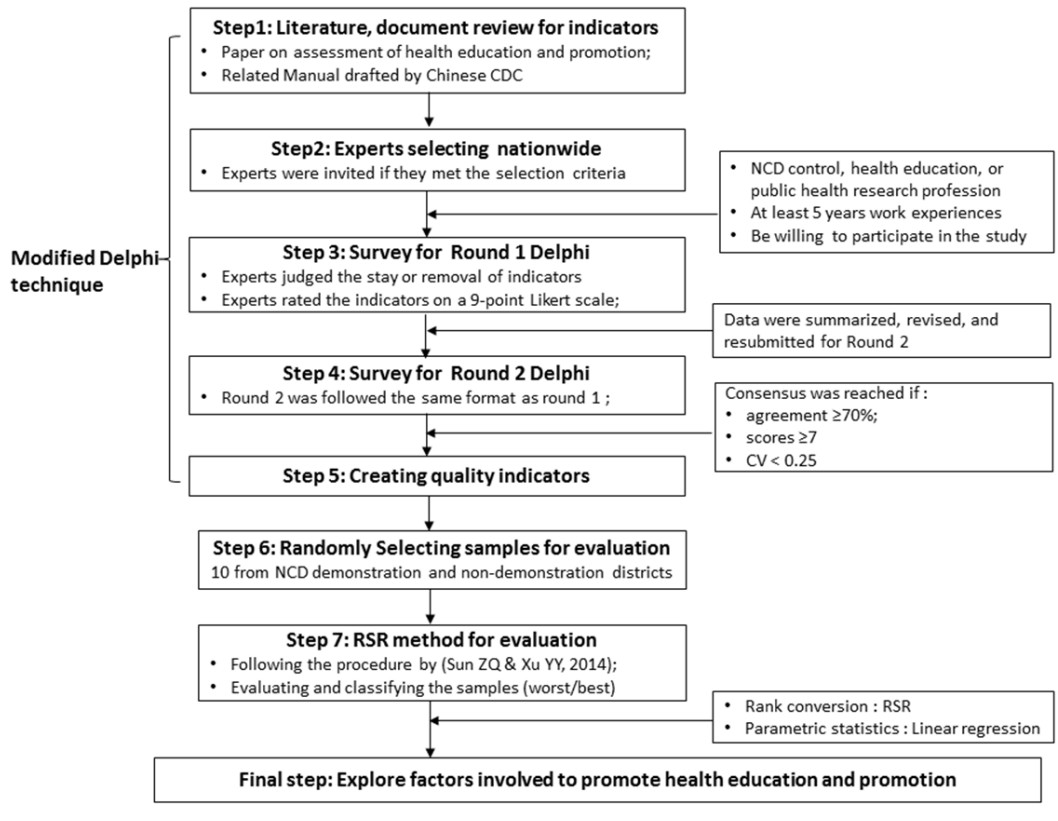

**Figure 1  The overall flow diagram of this study design.**

## Design for consultative indicators

By combining literature review with the related manuals of NCD demonstration districts drafted by the Chinese center for diseases control and prevention (CDC), we modified and designed a 38- consultative item for the Delphi process. These items were categorized into seven themes, including organizational management, financial support, professional personnel, health education and health promotion, heath awareness and behaviors of NCDs, management and control in NCD patients, others (such as satisfaction with supplies of health education and promotion, health literacy level) (Table S1).

## Selection of experts for Delphi survey

We sent invitations to more than 20 potential experts via e-mail or letter, in which we explained to experts the study background, gave a brief introduction of Delphi method, and what experts needed to do. In total, 19 experts responded and agreed to participate in the Delphi process. Experts across the country were selected for this study using the following criteria:

- Be working for the department NCD control and prevention of CDC, health education institution, or public health research (especially on NCDs) in university.
- Have had work experience for at least five years at provincial or national institutions.
- Be interested in participating in this study.

### Instrument for Delphi surveys

We listed the instrument for the Delphi surveys as the following four parts:

- General characteristics of the experts, including age, profession, professional title, education, years of working experiences and so on.
- A basic guideline for finishing the Delphi survey.
- Questionnaire with consultative indicators for evaluation of health education and promotion.
- Definition of experts' authority both in familiarity with indicators and judgment criteria for the indicators.

### Two-round Delphi process

With its anonymous consultation, feedback information, statistical inference, and easy utilization (*Holbech et al., 2017*; *Kuster et al., 2015*; *Gracht, 2012*), the Delphi technique is a popular assessment tool in health care research (*Zhao et al., 2015*; *Balaguer et al., 2016* *Jones & Hunter, 1995*). We conducted a two-round Delphi process between late 2013 and 2014 as follows: during round 1, the experts needed to judge whether the items should be included based on validity and feasibility and were free to make comments. They also rated each item for importance on a 9-point Likert scale (*Mrowietz et al., 2014*; *Suzuki et al., 2012*). The scores from 1 to 9 correlate to "strongly unimportant" to "strongly important", while 0 indicates disagreement.

The data from the round 1 were then summarized, revised, and resubmitted for round 2, following the same format as round 1 to determine quality indicators.

### Consensus definition

The expert agreement, average scores, and coefficient of variation (CV) are internationally taken as common parameters for determining consensus in the Delphi process. However, there are controversies regarding the parameter thresholds (*Hasson, Keeney & McKenna, 2010*; *Flores, Marshall & Cordina, 2014*; *Slade et al., 2014*; *Strosberg et al., 2015*; *Jang et al., 2015*). We integrated the parameters with a modified criterion to avoid important items from potentially being removed.

In the first round, consensus level was set as: (1) Expert agreement (%): items were removed with agreement $\leq 40\%$ and included when agreement $\geq 70\%$. They were considered uncertain when agreement ranged from 40% to 70%. (2) Median scores: items were excluded with scores $\leq 4$, included when scores $\geq 7$, and temporarily included when scores were between 4 and 6. (3) CV: items were included with CV $<0.25$. In the second round, consensus was reached only if agreement $\geq 70\%$, scores $\geq 7$, and CV $<0.25$.

### Selection of samples and RSR evaluation

After the Delphi process, ten districts from the Hunan province were selected and coded as samples for assessment with a simple randomizing function. This study conducted data collection from evaluated districts between 2014 and 2015 on schedule, when twenty-eight districts got the nomination in the province. Therefore, we randomly selected six districts or counties representing NCD demonstration districts. Meanwhile, we also randomly chose

four districts or counties from the same cities as that of NCD demonstration districts, as control subjects from the rest of the non-demonstration districts in the province.

Finally, the RSR method was conducted in 2015 to assess the performance of the sample districts. As a comprehensive evaluation method, the RSR was developed by Sun & Tian, who also proved its validation and rationality (*Sun & Tian, 1994*; *Wang et al., 2015b*). The basic theory of the RSR method is that a dimensionless statistical indicator (RSR) is calculated from an n × m matrix through rank conversion (*Wang et al., 2015b*). With the indicator RSR, a parametric statistic by the linear regression equation ($R\hat{S}R$) was followed to analyze the RSR distribution. The statuses (worst/best) of evaluated districts were evaluated or classified based on the $R\hat{S}R$. The RSR procedure was carried out here through the following steps:

The original values of items were converted to high-quality values. There was no conversion work here due to their natural high-quality characteristics.

- Quality indicators within the ten sample districts were ranked based on their original values. All indicators were ranked in ascending order as $R_{ij}$ ($i \leq n, j \leq m$)
- The RSR was calculated with the equation $RSR_i = \frac{\sum_{j=1}^{m} R_{ij}}{m \times n}$.
- The distribution of RSR was analyzed based on a Probit (-oriented downward accumulative frequency (p).
- A linear regression was set up. Taking $RSR_i$ as a dependent variable and $P$ as an independent variable separately, an equation was built using the formula as $R\hat{S}R = a + b \times$ Probit.
- With the $R\hat{S}R$, four levels (from best to worst) were classified into and set for the evaluation of the sample districts as follows: level I ($<P_{6.681}$, Probit $<$ 3.5), level II ($P_{6.681}$, Probit: 3.5 $\sim$), level III ($P_{50}$, Probit: 5 $\sim$), and level IV ($P_{93.319}$, Probit: 6.5 $\sim$) (*Sun & Xu, 2014*).

## Ethical approval

We received written, voluntary, informed consent when the completed questionnaires from the experts invited by us were submitted. This study was reviewed and approved by the Ethics Committee of Hunan Provincial Centre for Disease Control and Prevention, China (HNCDC/JL31-044: 2013011).

## Statistical analysis

Characteristics of experts were shown as mean ±SD (standard deviation) and frequency (percentage). Variables resulted from Delphi processes were reported as frequency (percentage), M (Median), Mode and CV. Student's $t$-test ($\alpha = 0.05$) was applied to test if both Kendall's W and Cronbach's $\alpha$ between the two round Delphi processes are significantly different from each other. In order to verify the $RSR_i$ in different districts which were statistically ranked and classified, an analysis of variance (ANOVA) was used to perform a hypothesis test ($\alpha = 0.05$) on the linear regression fitting.

Data was handled and described by Microsoft Excel 2010. The statistical analyses were performed with SPSS 22.0 version (IBM SPSS Inc., Chicago, IL, USA).

**Table 1  The description of experts participating in the study.**

| Items | Sub-items | N (n = 19) | % |
|---|---|---|---|
| Age (yr) | <40 | 4 | 21.05 |
| | 40–49 | 11 | 57.89 |
| | 50–59 | 3 | 15.79 |
| | ≥60 | 1 | 5.26 |
| professional title (public health) | [a]Chief doctor or professor | 10 | 52.63 |
| | [a]Associate chief doctor or associate professor | 7 | 36.84 |
| | Attending doctor | 2 | 10.53 |
| Academic degree | MD | 6 | 31.58 |
| | Masters | 6 | 31.58 |
| | Bachelors | 7 | 36.84 |
| Professional background | NCD control and prevention | 10 | 52.63 |
| | Health education and health promotion | 6 | 31.58 |
| | Public health | 3 | 15.79 |
| Years in current job | 5–9 | 3 | 15.79 |
| | 10–19 | 11 | 57.89 |
| | 20–29 | 4 | 21.05 |
| | ≥30 | 1 | 5.26 |

**Notes.**

MD, Doctor of medicine; NCD, Non-communicable disease.

[a]equivalent for both.

# RESULTS

## Results of the Delphi process

In total, 19 experts completed each question in both rounds of the Delphi survey. They were working either for NCD control and prevention departments of CDC, health education institutions, or schools of public health in universities. They averaged $44.84 \pm 6.69$ years old, with a mean of $21.37 \pm 7.90$ working years in their current job. In total, 89.5% (17/19) of the experts were recognized as senior doctors. All of the experts had a bachelor's degree of public health, and 63.2% (12/19) of them also had a master's degree (Table 1).

In the first round, five items were removed due to CV values ≥0.25, five other items temporarily remained either because of the agreement between 40% and 60%, or of scores between 4 and 6 (Table 2). Thirty-three items remained and were merged into thirty-one items in response to experts' opinions. In the second round, five items were removed either because of the agreement <70% or of median scores unqualified. Two items were merged into one item (Table 3). Finally, twenty-five items were included in the framework of the quality indicators, including the following themes: organizational management, financial support, professional personnel, health education and health promotion, residents' health awareness and behaviors of NCDs, residents' satisfaction with supplies from health education and health promotion, and the health literacy of the resident (Table 4).

**Table 2  Results from the round 1 Delphi process in this study.**

| Themes | Seq. | Agreement (%) | Median | Mode | CV |
|---|---|---|---|---|---|
| Organizational management | 1 | 100.00 | 9 | 9 | 0.11 |
| | 2 | 78.95 | 8 | 8 | 0.14 |
| | 3 | 100.00 | 8 | 9 | 0.18 |
| | 4[a] | 89.47 | 6 | 8 | 0.28 |
| Financial support | 5 | 100.00 | 9 | 9 | 0.07 |
| | 6 | 78.95 | 8 | 9 | 0.19 |
| | 7 | 89.47 | 8 | 8 | 0.15 |
| Professional personnel | 8 | 89.47 | 7 | 8 | 0.15 |
| | 9[a] | 68.42 | 6 | 6 | 0.30 |
| Health education and health promotion | 10 | 100.00 | 8 | 8 | 0.12 |
| | 11 | 100.00 | 8 | 8 | 0.12 |
| | 12 | 100.00 | 7 | 8 | 0.18 |
| | 13 | 94.74 | 7 | 7 | 0.19 |
| | 14 | 84.21 | 7 | 6 | 0.17 |
| | 15 | 84.21 | 7 | 6 | 0.16 |
| | 16 | 78.95 | 6 | 6 | 0.18 |
| | 17 | 78.95 | 7 | 7 | 0.17 |
| | 18 | 94.74 | 6 | 6 | 0.17 |
| | 19 | 68.42 | 6 | 5 | 0.23 |
| | 20 | 94.74 | 7 | 6 | 0.20 |
| | 21 | 89.47 | 7 | 6 | 0.19 |
| | 22 | 100.00 | 8 | 8 | 0.13 |
| | 23 | 100.00 | 7 | 7 | 0.23 |
| | 24 | 94.74 | 7 | 7 | 0.22 |
| | 25 | 94.74 | 7.5 | 8 | 0.15 |
| | 26 | 94.74 | 7.5 | 7 | 0.15 |
| | 27 | 47.37 | 6 | 6 | 0.24 |
| | 28 | 47.37 | 6 | 6 | 0.22 |
| Awareness and behavior of NCDs | 29 | 100.00 | 8 | 9 | 0.19 |
| | 30 | 68.42 | 8 | 9 | 0.21 |
| Management and control in NCD patients | 31[a] | 52.63 | 6.5 | 9 | 0.27 |
| | 32 | 68.42 | 8 | 9 | 0.19 |
| | 33[a] | 42.11 | 7 | 7 | 0.26 |
| | 34 | 52.63 | 7 | 7 | 0.18 |
| Other | 35 | 68.42 | 8 | 8 | 0.17 |
| | 36 | 94.74 | 7.5 | 9 | 0.24 |
| | 37[a] | 73.68 | 6 | 6 | 0.26 |
| | 38 | 100.00 | 7 | 8 | 0.23 |
| Total | – | – | 7 | 8 | 0.20 |

**Notes.**
  [a] Items removed from the process.
   CV, Coefficient of variation.

**Table 3  Results from the round 2 Delphi process in this study.**

| Themes | Seq. | Agreement (%) | Median | Mode | CV |
|---|---|---|---|---|---|
| Organizational management | 1 | 100.00 | 9 | 9 | 0.15 |
| | 2 | 73.68 | 8 | 8 | 0.20 |
| | 3 | 100.00 | 8 | 9 | 0.15 |
| | 4[a] | – | – | – | – |
| Financial support | 5 | 100.00 | 9 | 9 | 0.05 |
| | 6 | 89.47 | 8 | 9 | 0.18 |
| | 7 | 89.47 | 8 | 8 | 0.11 |
| Professional personnel | 8 | 94.74 | 8 | 8 | 0.12 |
| | 9[a] | – | – | – | – |
| Health education and health promotion | 10 | 100.00 | 8 | 8 | 0.09 |
| | 11 | 100.00 | 8 | 8 | 0.09 |
| | 12 | 100.00 | 7 | 8 | 0.18 |
| | 13[b] | – | – | – | – |
| | 14 | 84.21 | 7 | 7 | 0.16 |
| | 15[c] | 78.95 | 6 | 7 | 0.16 |
| | 16 | 78.95 | 7 | 7 | 0.16 |
| | 17 | 94.74 | 7 | 6 | 0.15 |
| | 18[c] | 89.47 | 6 | 6 | 0.17 |
| | 19[b] | – | – | – | – |
| | 20 | 94.74 | 7 | 7 | 0.16 |
| | 21 | 89.47 | 7 | 7 | 0.20 |
| | 22 | 100.00 | 8 | 8 | 0.09 |
| | 23 | 94.74 | 7 | 7 | 0.18 |
| | 24 | 89.47 | 7 | 7 | 0.16 |
| | 25 | 100.00 | 8 | 9 | 0.14 |
| | 26 | 94.74 | 8 | 9 | 0.13 |
| | 27[c] | 57.89 | 7 | 7 | 0.22 |
| | 28[c] | 52.63 | 7 | 7 | 0.22 |
| Awareness and behavior of NCDs | 29 | 100.00 | 8 | 8 | 0.08 |
| | 30 | 73.68 | 8 | 8 | 0.17 |
| Management and control in NCD patients | 31[a] | – | – | – | – |
| | 32[c] | 68.42 | 7 | 7 | 0.20 |
| | 33[a] | – | – | – | – |
| | 34[c] | 47.37 | 6 | 6 | 0.12 |
| Other | 35 | 78.95 | 7 | 7 | 0.18 |
| | 36 | 100.00 | 7 | 7 | 0.15 |
| | 37[a] | – | – | – | – |
| | 38 | 94.74 | 8 | 8 | 0.14 |
| Total | – | – | 8 | 8 | 0.16 |

**Notes.**
[a] Items removed from round 1.
[b] Items merged with the prior item.
[c] Items removed from round 2.
CV, Coefficient of variation

**Table 4  Framework of quality indicators for evaluation of health education and promotion.**

| Subject items | Sub-subject items | Code |
|---|---|---|
| Organizational management | Local government based leadership team on NCDs control was established and held meetings once at least per year | 1 |
| | Local health authority based leadership team on NCDs control was established and held meetings once at least per year | 2 |
| | whether a yearly work plan on health education and promotion of NCDs was made | 3 |
| | the number of NCDs special fund by local government per thousand population per year (RMB, yuan) | 4 |
| Financial support | the number of NCDs control expenditures in local CDC (ten thousand yuan) | 5 |
| | the proportion of NCDs control expenditures in total business expenses in local CDC (%) | 6 |
| Professional personnel | the number of persons in NCDs health education and promotion institutions beyond village level per thousand population | 7 |
| | whether a yearly NCDs related health broadcasting planning was developed | 8 |
| | Whether billboards on NCDs control were presented and advertised regularly in local medias (except TV) | 9 |
| | the frequencies and average minutes (per time )of promotion on NCDs control and prevention in local TV station per year | 10 |
| | the mean sorts of materials printed and promotion billboards of NCDs control and prevention | 11 |
| | the mean sorts of NCDs control and prevention video presented by town level hospitals | 12 |
| | the times of public consultation of NCDs related core information on different themes per year | 13 |
| | the community based coverage of NCDs control and prevention billboard (%) | 14 |
| Health education and health promotion | the average monthly frequencies of NCDs control and prevention billboard updating in community | 15 |
| | the average coverage of fitness center or room in community (%) | 16 |
| | the times of NCDs related health lecture in community (a scale of >50 persons) | 17 |
| | the times of massive promotion activities of NCDs per year(a scale of >100 persons) | 18 |
| | the institution based coverage of NCDs control lectures in both elementary and middle school (%) | 19 |
| | the students based coverage of NCDs control lectures in both elementary and middle school (%) | 20 |
| Heath awareness and behaviors of NCDs | people's awareness rate of NCDs control and prevention (%) | 21 |
| | the rate of people's healthy behavior formation(%) | 22 |
| | Whether the assessments of NCDs risk factors had been conducted during the past 3 years | 23 |
| Others | people's satisfaction with supplies of health education and promotion | 24 |
| | people's health literacy level in NCDs control and prevention | 25 |

The Kendall's W was 0.35 in round 1 and 0.45 in round 2, with a significant difference ($P < .001$), showing that no further rounds were needed due to adequate item agreement among the experts. Meanwhile, we used Cronbach's $\alpha$ to measure the internal consistency of indicators. Our study showed that Cronbach's $\alpha$ was 0.90 in round 1 and 0.85 in round 2, with a 95% confidence interval (CI) of 0.82–0.95 and 0.74–0.93, respectively, showing a good internal constancy of indicators (*Benhamou et al., 2013*; *Bland & Altman, 1997*).

**Table 5 The result from the evaluated districts with the Rank Sum Ratio method.**

| Code | RSR | $f$ | $\bar{R}$ | $p$ | Probit | $R\hat{S}R$ | Classification[a] |
|------|------|-----|-----|------|--------|-------|-------------------|
| A | 0.8173 | 1 | 10 | 97.5 | 6.96 | 0.8687 | A |
| F | 0.7577 | 1 | 9 | 90 | 6.28 | 0.7442 | B |
| B, C | 0.6635 | 2 | 7.5 | 75 | 5.67 | 0.6326 | B |
| E | 0.6192 | 1 | 6 | 60 | 5.25 | 0.5558 | B |
| D | 0.6019 | 1 | 5 | 50 | 5.00 | 0.5100 | B |
| H | 0.3846 | 1 | 4 | 40 | 4.75 | 0.4643 | C |
| I | 0.3558 | 1 | 3 | 30 | 4.48 | 0.4148 | C |
| J | 0.3365 | 1 | 2 | 20 | 4.16 | 0.3563 | C |
| G | 0.2904 | 1 | 1 | 10 | 3.72 | 0.2758 | C |

**Notes.**
RSR, Rank Sum Ratio
[a] A to C: best-ranked to third-best ranked.

## Results of sample selection

Ten districts or counties were chosen as follows: six NCD demonstration districts, including one national-nominated demonstration district, namely Furong District (A). Five provincial-nominated demonstration districts, namely Ziyang District (B), Shaodong County (C), Shuangfeng County (D), Luxi County (E), and Yuhua District (F). There were four non-NCD demonstration districts: Anhua County (G), Xinhua County (H), Xinshao County (I), and Jishou County (J). The original values of twenty-five quality indicators in these districts were collected from field investigations; those are shown in Table S2.

## Results of RSR

The RSR procedure was followed to assess and classify the sample districts through both rank conversion and parametric statistics. It showed that the ten sample districts were classified into four levels (from best to worst). The Furong District was at the best level, with the highest $R\hat{S}R$ of 0.8687. Five NCD demonstration districts were at the second-best level with $R\hat{S}R$ ranging from 0.5100 to 0.7442: Ziyang District, Shaodong County, Shuangfeng County, Lu xi County, and Yuhua District. All four non-demonstration districts were at the third-best level with the $R\hat{S}R$ r anging from 0.2758 to 0.4643: Anhua County, Xinhua County, Xinshao County, and Jishou County. None of the districts were at the fourth-qualified level (Table 5).

## DISCUSSION

We comprehensively used two assessment tools by combining the modified Delphi and RSR technique to offset the limitations of a single assessment tool. The modified Delphi method conducted in our study included richer information than the traditional Delphi method. Additionally, it is more flexible in the number of rounds, as the process is closed once a consensus is reached among experts (*Van Vliet et al., 2016*), thus avoiding possible redundant work. The RSR method is a comprehensive evaluation tool for multi-indicators with the advantages of having no data type restrictions or bias of abnormal values (*Wang et al., 2015b*; *Sun & Xu, 2014*; *Wang et al. 2015a*).

This study showed that the ten sample districts belonged to the first three levels of health education and health promotion, with none at the fourth-qualified level. This indicated that a negative situation for health education and health promotion did not exist in Hunan province.

Overall, NCD demonstration districts performed better in health education and health promotion than non-NCD demonstration districts. The first contributing factor to better performances in the NCD demonstration districts was organizational management. All NCD demonstration districts had a local government and/or health authority-based leadership team on NCD control, compared to zero teams in non-demonstration districts. This was related to the nature of the programs of NCD demonstration districts in China. The programs were run by the local governments of each district in China (*The National Health and Family Planning Commission of China, 2016*). The support of the local governments was crucial to the success of the program. Their attention shows not only in setting up a leadership team but often in providing more financial support, another factor differing greatly between NCD demonstration districts and non-demonstration districts in this study. Whether it is the amount of NCD special funds from the government or the proportion of NCD expenditures in total expenses in CDC, the financial support in NCD demonstration districts greatly exceeded those of non-NCD demonstration districts in which no or very few funds were available.

The essence of the theme of health education and health promotion in NCD demonstration districts also obviously surpassed that of non-demonstration districts. This was reflected in their development of health broadcasting planning, periodic promotions of NCD control recommendations in media, public consultation of NCD core information, the frequency of updating NCD control recommendations on bulletin boards in the community, and community-based coverage of fitness centers. Actions such as systematic and frequent media propaganda and community-based supportive environment were often canceled or poorly conducted due to insufficient funds. The NCD demonstration districts in this study (except Luxi County) all had more economic resources than non-NCD districts, and three of these NCD demonstration districts (50%) were urban-level districts. Although Luxi is a national poverty-stricken county (*The State Council Leading Group Office of Poverty Alleviation and Development of China & The National Development and Reform Commission, 2011*), it still received a special fund from the local government every year because of the NCD demonstration district program. Meanwhile, the four control subjects were all rural-level districts, 75% of which were national poverty-stricken counties (*The State Council Leading Group Office of Poverty Alleviation and Development of China & The National Development and Reform Commission, 2011*). Another contributing factor is the lower focus on NCD control and prevention in China. Compared to other public health problems such as infectious diseases and public health emergencies, NCD control is still inadequately addressed across the country despite the increased attention it has received (*Chen, 2015*; *USA Centers for Disease Control and Prevention, 2015*). Thus, even though a lot of health education and promotion work (for example technical support and community-based promotion), has been conducted in some districts, there has been little focus on NCD control and prevention. Additionally, the lack of a reward and punishment

system based on systematic evaluation was also a possible factor contributing to the differences.

An interesting result from this study was that although both the people's awareness rate and health literacy of NCD control and prevention were higher in NCD demonstration districts than in non-demonstration districts, the rate of people's healthy behavior towards NCD control in demonstration districts had a narrow difference from that of non-demonstration districts. This result indicated that there is still a long way to go from people's awareness to healthy behavior.

The six demonstration districts were still at two levels. Furong District was the only subject at the best level with many best-performing indicators, especially in terms of financial support and technical support for NCD-related promotion materials, and people's indicators of NCD control, such as health awareness, healthy behaviors, health literacy, and satisfaction with supplies of health education and promotion. Its level of performance benefitted from advantages both in the form of a robust economy (i.e., downtown in the capital city of Hunan) and a historically strong foundation in health education and health promotion. It is worth discussing the second advantage: Furong District had launched a famous "Ten Health" project before the program of NCD demonstration district. The "Ten Health" project was rich in content, including overall health mobilization, massive health lecture, etc. The district had also built a community-based NCD control database and information sharing model. These actions indeed benefitted the residents. This could explain the best-performance of the aforementioned people's health indicators in the Furong District.

### Limitations

One limitation of this study was that the number of evaluated samples (both NCD demonstration districts and non-demonstration districts) was inadequate due to some restrictions such as insufficient funding and that few NCD demonstration districts available during the study period. This might fail to fully represent the whole status of Hunan. Another limitation was that the assessment in this study was based on retrospective data of the evaluated samples, thus, it failed to reflect on the real-time situation of the sample districts, or account for the potential for more non-demonstration districts to join the program of NCD demonstration districts in the future.

## CONCLUSIONS

With the integrated Delphi and RSR method, we assessed health education and health promotion work, which is an important part of the NCD demonstration district program in China. We comprehensively and comparatively assessed their performance both in NCD demonstration districts and non-demonstration districts. Our study showed that NCD demonstration districts performed better in health education and health promotion work than non-demonstration districts. To promote this work, emphasis should be placed on the organizational management in government, financial support, media-related health

broadcasting and promotion, publicity materials-based technical support, community-based health promotion and supportive environments, and people's enhanced awareness and health literacy of NCD control.

## ACKNOWLEDGEMENTS

We thank all experts for their participation in the Delphi process. We also extend our thanks to the following CDCs for their assistance in data collection and survey conduction: Furong District CDC, Yuhua District CDC, Ziyang District CDC, Shaodong County CDC, Shuang feng County CDC, Luxi County CDC, Anhua County CDC, Xinhua County CDC, Xinshao County CDC, and Jishou County. Both Dr. Zundong Yin (MD in Epidemiology and Biostatistics) from the Department of National Immunization Program of the Chinese CDC, and Dr. Fuqiang Liu (MD in Epidemiology and Biostatistics) from Hunan Provincial CDC were highly appreciated for their suggestions on this paper.

### Funding

This work was supported by China Hunan Provincial Science & Technology Department (No. 2013zk2054). The funders had no role in study design, data collection and analysis, decision to publish, or preparation of the manuscript.

### Grant Disclosures

The following grant information was disclosed by the authors:
China Hunan Provincial Science & Technology Department: 2013zk2054.

### Competing Interests

The authors declare there are no competing interests.

### Author Contributions

- Qiaohua Xu conceived and designed the experiments, performed the experiments, analyzed the data, contributed reagents/materials/analysis tools, prepared figures and/or tables, authored or reviewed drafts of the paper, approved the final draft.
- Biyun Chen performed the experiments, contributed reagents/materials/analysis tools, approved the final draft.
- Donghui Jin conceived and designed the experiments, performed the experiments, contributed reagents/materials/analysis tools, approved the final draft.
- Li Yin contributed reagents/materials/analysis tools, prepared figures and/or tables, approved the final draft.
- Yuelong Huang conceived and designed the experiments, approved the final draft, project declaring, management of project.

### Human Ethics

The following information was supplied relating to ethical approvals (i.e., approving body and any reference numbers):

We received written, voluntary, informed consent when the completed questionnaires from the experts invited by us were submitted. This study was reviewed and approved by the Ethics Committee of Hunan Provincial Centre for Disease Control and Prevention, China (HNCDC/JL31-044: 2013011).

## Data Availability

The raw measurements are available in the Supplemental Files.

## Supplemental Information

Supplemental information for this article can be found online at http://dx.doi.org/10.7717/peerj.6579#supplemental-information.

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
