# Peer review of "An assessment for health education and health promotion in chronic disease demonstration districts: a comparative study from Hunan Province, China"

_PeerJ, doi:10.7717/peerj.6579_

## Round 0.1 · original submission · Minor Revisions

Please be attentive to the comments of the reviewers and attend to all of them where possible. This editor takes issue with the one reviewer's comment that a large enough sample size is not used. If that were the case the reviewer should have rejected the manuscript on first review because collecting more observations is not feasible.

The most salient issue brought out by the reviews is the need for native English language editing. Please have a native English speaker edit the paper or pay a service to edit the paper. Thank you very much for your efforts. I truly look forward to receiving a "clean" copy of your work.

Reviewer 1 ·

Basic reporting

• English language: The English language should be improved to make sure that the international audience can understand the text. Think about having a native English speaking colleague review the manuscript. Some examples where the language could be improved include lines 28, 29, 42-43, 44, 71-72,…
• There is clearly a need for the creation of quality indicators to measure health education and promotion efforts. Potentially think about working on flow of introduction to help the author better grasp the need.
• Professional article structure, figs, tables: I am not familiar with RSR, so I am only commenting on Delphi-related tables and figures. Potentially think about expanding Figure 1 to visual show what occurred during Round 1 and Round 2 of Delphi method. You have the potential to show a lot of information in an aesthetically pleasing manner that will be more easily understood versus the text. See Li e al. (2014) “Framework of behavioral indicators for outcome evaluation of TB health promotion: a Delphi study of TB suspect and TB patients” for their Figure 1. Great article to use as a resource.
• Raw data shared and accessible.
• The article is a bit busy with three primary purposes.

Experimental design

• Primary research within Aims & Scope of journal: This article is focused on Health Sciences – mainly monitoring population health through a novel method.
• Research question well defined, relevant & meaningful: the research question could be improved and written more clearly. Abstract: “This study was designed to construct such quality indicators, have a grasp of the work status of health education and promotion in the districts, and explore the main factors involved to promote the work.” This is a bit busy, potentially think about splitting up. “This study was designed to establish a framework of indicators for outcome evaluation of health education and promotion efforts in Chinese districts.” And then continuing with your secondary purposes.
• Rigorous investigation performed to a high technical & ethical standard: Lines 131-132 – typically informed consent is a bit more formal than “it was regarded as informed consent.” (I am not an expert on RSR and therefore am not commenting.)
• Methods described with sufficient detail & information to replicate: improvement needed. More information about the Selection of Delphi experts, Instrument for Delphi surveys, consider putting the consensus level in bullet form for easy reading, Figure 1 could be improved (see Basic Reporting section), consider splitting Table 2 into two separate tables for ease of reading with 1st round results in one table, and 2nd round results in the second table.
• Creating a table devoted to the framework of indicators would be helpful to readers to cleaning depict your results.

Validity of the findings

• Again, I’m not speaking on RSR only Delphi method. Reporting the median and standard deviation is not what I am accustomed to, typically mean and standard deviation or median and mode. Would consider changing standard deviation to mode in Table 2.
• I suggest having a native English speaker assist in editing the discussion and conclusions to help make the sections more easily understandable and readable for your audience.

Additional comments

I think the creation of the quality indicators by the modified-Delphi process should be the main focus of the paper. More information could be provided on the process while lessening the focus on the descriptives of the counties and RSR.

Reviewer 2 ·

Basic reporting

English is hard to follow with adjectives used to define results without any quantitative parameter.

Experimental design

Research question well defined, relevant & meaningful. Rigorous investigation is not performed.

Validity of the findings

Data is not controlled.

Additional comments

The number of participants in this study is too small. It is not possible to reach any conclusion with this number. There are other problems. The manuscript is hard to follow, and its fundamental hypothesis is not underlined. Abstract is written very poorly and does not provide clear understanding of results. The authors did not define DELPHI process in Introduction or results. The authors use adjective to define their results however how that conclusion is reached is not addressed.
1. The study participants are senior doctors with a mean of 21 year experience. What is the control of this group, a comparison of individuals with similar work experience who are not doctors. The other group of individuals who are doctors but have significantly less experience is required to compare with first group.
2. Line 23 “This study was designed to construct such quality indicators, have a grasp of the work status of health education and promotion in the districts, and explore the main factors involved to promote the work. “ The authors did not define quality indicators.
3. Line 82 “A modified Delphi method (conducted between late 2013 and 2014) was firstly used to build quality indicators. “ What is quality indicators?
4. Line 204 “Overall, NCD demonstration districts performed better in health education and health promotion than that of non-NCD demonstration districts. “ Define better ?
5. The English language needs to be improved. Many sentences are too long and are vague.

---

## Round 0.2 · accepted · Accept

Thank you very much for addressing all the reviewers' comments. Thank you especially for the language editing. Congratulations!

Reviewer 2 ·

Basic reporting

The English language is improved.

Experimental design

The experimental design is sound.

Validity of the findings

The study has interesting findings although validity remains unknown due to low number of samples.

Additional comments

I do not have additional comments. I will like to congratulate authors for their work.